# Strategies for Vaccine Prioritization and Mass Dispensing

**DOI:** 10.3390/vaccines9050506

**Published:** 2021-05-14

**Authors:** Eva K. Lee, Zhuonan L. Li, Yifan K. Liu, James LeDuc

**Affiliations:** 1NSF-Whitaker Center for Operations Research in Medicine and HealthCare, Georgia Institute of Technology, Atlanta, GA 30332, USA; leon.li@gatech.edu (Z.L.L.); yifanliu@gatech.edu (Y.K.L.); 2Galveston National Laboratory, University of Texas Medical Branch, Galveston, TX 77550, USA; jwleduc@utmb.edu

**Keywords:** vaccine prioritization, mixed vaccination strategy, switch trigger, mass vaccination, vaccine efficacy, biological-behavior-logistics-queueing computational framework

## Abstract

We propose a system that helps decision makers during a pandemic find, in real time, the mass vaccination strategies that best utilize limited medical resources to achieve fast containments and population protection. Our general-purpose framework integrates into a single computational platform a multi-purpose compartmental disease propagation model, a human behavior network, a resource logistics model, and a stochastic queueing model for vaccination operations. We apply the modeling framework to the current COVID-19 pandemic and derive an optimal trigger for switching from a prioritized vaccination strategy to a non-prioritized strategy so as to minimize the overall attack rate and mortality rate. When vaccine supply is limited, such a mixed vaccination strategy is broadly effective. Our analysis suggests that delays in vaccine supply and inefficiencies in vaccination delivery can substantially impede the containment effort. Employing an optimal mixed strategy can significantly reduce the attack and mortality rates. The more infectious the virus, the earlier it helps to open the vaccine to the public. As vaccine efficacy decreases, the attack and mortality rates rapidly increase by multiples; this highlights the importance of early vaccination to reduce spreading as quickly as possible to lower the chances for further mutations to evolve and to reduce the excessive healthcare burden. To maximize the protective effect of available vaccines, of equal importance are determining the optimal mixed strategy and implementing effective on-the-ground dispensing. The optimal mixed strategy is quite robust against variations in model parameters and can be implemented readily in practice. Studies with our holistic modeling framework strongly support the urgent need for early vaccination in combating the COVID-19 pandemic. Our framework permits rapid custom modeling in practice. Additionally, it is generalizable for different types of infectious disease outbreaks, whereby a user may determine for a given type the effects of different interventions including the optimal switch trigger.

## 1. Introduction

Vaccination is one of the cornerstones in controlling infectious disease outbreaks, reducing mortality, and protecting population health [1,2]. However, the development and manufacturing of vaccines usually take months to years, and vaccine shortages during epidemic outbreaks are not uncommon [3,4,5]. While the U.S. government’s Operation Warp Speed program set an unprecedented landmark in the rapid development of highly viable vaccines against SARS-CoV-2, the virus that causes COVID-19, the manufacturing cycle and supply-chain limitations prevent rapid production at the scale needed to protect the global population in a timely manner. Similarly, the 2017 yellow fever outbreaks in Brazil and multiple African countries brought to light the insufficiency of vaccine supply [6,7], and exposed the intrinsic difficulty in controlling an outbreak without an adequate supply of vaccines. Even if vaccine supplies are available, planning and stockpiling vaccines for an entire nation is financially daunting and logistically complicated [8]. Resource-limited and inadequate supply chain systems in some developing countries further impede the distribution of vaccines [9]. Because of limited vaccine supply, the effect of reduced and diluted dosage has been studied [10,11,12], aiming to cover a larger population with an existing vaccine supply. Clinically, individual-level vaccine immunogenicity prediction using system biology and machine learning helps to personalize and target vaccine delivery only to those who benefit from it, thus avoiding potential adverse effects and preventing waste [13,14,15,16].

In addition to the effort to best utilize vaccines at individual-level, systematic prioritized vaccination is widely considered one of the best strategies to contain a pandemic [17,18]. Normally, healthcare workers [19], children, pregnant women, and individuals with special medical conditions will be classified as the high-risk groups and be the first to receive the vaccines. Proper definition of high-risk groups who should receive vaccines first has been extensively investigated [20,21]. Since searching for an optimal prioritized strategy for a general definition of risk groups under practical constraints can be computationally difficult, some studies solved a relaxation problem by assuming adequate vaccine supply [22] or prioritizing a relatively simply-structured risk group [23]. Other works on prioritized vaccination strategy include fusing the spatial information with demographic data [24], minimizing the years of life lost [25], and optimizing multiple evaluation metrics [26].

Wallinga et al. studied the optimal allocation of medical resources with limited data by targeting intervention measures at the group with the highest risk of infection per individual [27]. Mylius et al. studied the vaccination strategy under different supply scenarios of influenza vaccines and suggested prioritizing individuals with high-risk of complications [28]. Longini et al. studied the optimal vaccine distribution pattern based on different strains of virus, availability of vaccines, and structure of objective function [29]. Patel et al. used stochastic epidemic simulations and meta-heuristics to find the optimal vaccine distributions [30]. Computer simulation is also used to compare different prioritization strategies and definitions of risk groups to assist decision-making, as discussed in Lee et al. [31,32]. While most studies assume the supply of vaccines is instant, Meyers et al. optimized the allocation of vaccines when there is a delay in vaccine supply [33]. In practice, most public health departments in the U.S. use a mixed strategy approach by prioritizing the vaccines to cover the high-risk population initially, and switch to the general public at a later time. Lee et al. proposed a compartmental-based model to determine the optimal switch trigger between prioritized and non-prioritized strategies [34] and solved the problem of determining the switch point via numerical optimization.

In this study, we focus on a mixed vaccination strategy to mitigate the current COVID-19 pandemic. Since the first four cases reported in Wuhan on 29 December 2019, the COVID-19 pandemic has spread to 191 countries, infecting over 106.6 million people, and causing over 2.33 million deaths (9 February 2021). With the emergence of more contagious variants from United Kingdom, Brazil, South Africa, and Denmark, public health officials are racing against the clock to vaccinate as many people as possible. Several analyses on vaccine allocation based on risk, equity and age distribution have been proposed. The National Academies analyzed the framework for equitable allocation [35]. Some models partition the population according to age-groups [36], while others incorporate demographic information, and risk factors [37,38]. Although serological tests have been used as surrogates to determine the degree of exposure to the SARS-CoV-2 virus in a population, most of these tests do not measure the neutralizing antibodies, thus some seropositive individuals may remain susceptible to COVID-19 and could require protection via vaccination.

We develop a general-purpose modeling framework that incorporates disease spread with interventions to analyze the effect of mass vaccination on disease containment. A unique feature in our model is that the vaccination operations inside the dispensing clinics or point-of-dispensing sites (POD) are explicitly modeled. Similar to our work on human influenza [34], mass vaccination strategy is optimized using the RealOpt system [39,40,41]. We employed numerical optimization to determine an optimal switch trigger that transitions from a prioritized to a non-prioritized strategy that minimizes overall attack and mortality rates. We investigated the results under different vaccine supply levels and severity of the outbreak, as represented by the basic reproduction number and the initial infectious population. We used an agent-based simulation-optimization system within RealOpt (Center for Operations Research in Medicine and Healthcare, Atlanta, GA, USA) to validate our model results. We then performed sensitivity analyses and examined the robustness of the optimized mixed strategy.

Our analysis across hundreds of cities/counties reveals that employing an optimal mixed strategy can significantly reduce the attack and mortality rates, on average by 3-fold and 2.8-fold respectively, when compared to the full prioritized vaccination strategy. The findings suggest that delays in vaccine supply and inefficiencies in vaccination delivery can substantially impede the containment effort. A 3-week delay increases the total infection and mortality by nearly 3-fold on average and lengthens the containment period by 90%. The more infectious the virus, the earlier it helps to open the vaccine to the public. As vaccine efficacy decreases, the attack and mortality rates rapidly increase; this highlights the importance of early vaccination to reduce spreading as quickly as possible to lower the chances for further mutations to evolve and to reduce the excessive healthcare burden. To maximize the protective effect of available vaccines, of equal importance are determining the optimal mixed strategy and implementing effective on-the-ground dispensing. The optimal mixed strategy is quite robust against variations in model parameters and can be implemented readily in practice. Studies with our holistic modeling framework strongly support the urgent need for early vaccination in combating the COVID-19 pandemic.

The modeling framework not only applies to mitigating the current COVID-19 outbreaks, but can also be readily tailored to a specific infectious disease using the appropriate compartmental disease model. The system is attractive in practice since it can assist the decision-making process in real-time to achieve the maximum population protection. Public health responders and decision makers can input multiple model parameters with respect to the biological characteristics of the disease, vaccine supply, demographic information, health risk and other factors, and implement the resulting optimal switch trigger to facilitate vaccine dispensing in emergencies to minimize the overall infections and mortalities.

## 2. Materials and Methods

To study the transmission of COVID-19 in the population, we develop a baseline disease model with 8 compartments: (S) usceptible, (E) xposed, (P) Infectious, (A) symptomatic, (I) symptomatic, (Q) Recovered but still infectious, (R) ecovered, (D) eceased [42]. In addition to the transition of compartments due to disease propagation and new infections, the vaccination process is also modeled in detail to reflect the importance of vaccination in containing the COVID-19 outbreaks. Two more compartments are also added: Hospitalization: individuals with symptoms of COVID-19 (positive molecular test) and are seriously ill will be properly treated and quarantined, and Vaccinated and immune: individuals who receive vaccines and develop immunity. We separate the population into two environments: (a) intra-POD: those who are currently at point-of-dispensing sites (POD) to receive COVID-19 vaccine and (b) outer-POD: the outside general public. This classification is intuitive and effective since individuals in these two environments have different contact rates.

At the POD, besides the potential natural propagation of COVID-19 by undetected infectious individuals, the layout of the facility and its operations efficiency in triage, vaccination and management of crowds also play a critical role in mitigating intra-facility disease transmission. A typical POD consists of at least three stations: at the triage/screening station providers determine if an individual is suitable to receive vaccine. At the medical counseling station providers advise individuals who may have already been infected about self-quarantining or seeking medical treatment, and answer questions and address concerns regarding the vaccine. At the vaccination station providers administer the vaccines. Upon vaccination, individuals will be observed for a period of 15–30 min to ensure they do not experience acute adverse reactions from the vaccine [43].

The operations inside the PODs are modeled as time-variant parameters in the disease propagation schema in Figure 1. Figure 2a shows a POD setup specifically for COVID-19 vaccine dispensing. Figure 2b illustrates the disease dynamics of people and their interplay within and between stations at a dispensing site.

Although online/phone scheduling and “screening” is performed to ensure only those eligible are scheduled for vaccination, individuals at various stages—susceptible, exposed, pre-symptomatic infectious, and asymptomatic—may arrive at the POD (Figure 3, first column light blue arrows). We assume that only the susceptible and exposed individuals may develop immunity after vaccination according to an estimated vaccine efficacy probability (Figure 3, deep green arrows). Figure 3 shows some individuals may become exposed to the disease during the vaccination process (red dashed arrow). Once vaccinated, it typically takes a few weeks after vaccination for the body to build protection (immunity) against the virus that causes COVID-19. That means it is possible a person could still get COVID-19 just after vaccination (the light-blue disease progression). Some vaccinated people may develop only partial immunity and may contract disease still, although mostly with only mild or no symptoms at all (dark blue disease progression). A small percentage may fail to develop immunity.

The general-purpose modeling framework and the systems of ordinary differential equations (ODEs) and their application to COVID-19 are briefly described in the Appendix A.

To demonstrate the importance of prioritizing vaccination, we partition the population into groups based on age, service, and health-risk: “healthy individuals”, “healthcare workers”, “elderly 65 years-old or above”, and “patients under 65 with high-risk health conditions.” The latter three groups are categorized as high-risk. Healthcare workers have higher contact rates to infections (41% to 63%) [44,45]. Elderly 75-year-old or above [46,47], and patients with associated health conditions are more vulnerable to infection and mortality [48]. The Centers for Disease Control and Prevention estimate that over 85% of older adults (65+ years old) have at least one chronic disease [49]. We will use the risk group model with these groups to discuss the prioritization of vaccines. There is currently no approved vaccine formula for those under 16 years-old, though Moderna’s mRNA-1273 clinical trials for 12- to 17-year-olds started in late December 2020 in the United States [50], and United Kingdom is planning for a clinical trial of Oxford-AstraZeneca coronavirus vaccine AZD1222, for pediatric ages 6–17 [51].

To realistically reflect the logistics in emergency response, we assume that the arrival rate at the PODs is a time-variant parameter instead of uniform throughout the duration of the disease outbreak. Let λ denote the basic pooled arrival rate of individuals, i.e., it is the arrival rate at the beginning of the vaccination process when non-prioritized policy is used. If prioritized policy is used, all the arrivals are from the high-risk group, with population adjusted arrival rate $\lambda \left(t\right)=\lambda \frac{{N_{0}}^{\prime }\left(t\right)}{N^{\prime }}$ where $N^{\prime }$ is the total population of high-risk group, and ${N_{0}}^{\prime }\left(t\right)$. is the high-risk population at time *t* who have not yet visited the vaccination facility. As vaccination progresses, the arrival rate at the PODs will gradually decrease as more individuals are vaccinated or treated. When vaccines are open to the public (non-prioritized policy), the arrival rate will be expressed as $\lambda \left(t\right)=\lambda \frac{N_{0}\left(t\right)}{N}$. where *N* and $N_{0}\left(t\right)$ are the initial population and the population who have not yet visited POD from both the high-risk and normal groups. The basic arrival rate λ is inferred from our previous mass vaccination time-motion studies [39,52], and the composition of population is obtained using demographic statistics. Based on the expression of the arrival rate, there will be an increase in the arrival rate upon the switching, since the high-risk population is significantly smaller compared to the normal healthy adult group.

A *mixed strategy* is a vaccination approach that first prioritizes the vaccines to cover the high-risk population, and switches to the general public at a later time. The switch trigger (associated with a mixed strategy), *g*, is expressed as the percentage of vaccine given to the high-risk population before vaccination is opened to the general population. Let π be the percentage of high-risk population, and *c* denote the vaccine inventory. When the switch occurs, gc/π of the high-risk population are vaccinated. When g=0, the corresponding strategy is *full nonprioritized*, i.e., vaccines are administrated to everyone with no special priority given to the high-risk groups. When $g=\min\{\frac{\pi }{c},\;1\}$ the associated strategy is *full prioritized* where all vaccines are distributed first to high-risk groups.

We use the overall attack rate and mortality rate as evaluation metrics. Overall attack rate measures the cumulative percentage of population infected with COVID-19 during the course of the outbreak. We seek an optimal switch trigger $g^{*}$. for a mixed strategy that minimizes the overall attack and mortality rates by the end of the outbreak, with the constraint on availability of vaccine inventory. In our analysis, we contrast continuous vaccine supply levels versus batch arrival. Our earlier study showed that the overall protective effect of batched vaccine supply is far inferior due to the delay in vaccine supply and/or in dispensing, rendering a higher overall infectivity and mortality [34]. The two objectives, attack and mortality rates, are derived from the output of the ODE systems, and are controlled by the switch trigger $g^{}$. Therefore, to obtain an optimal $g^{*}$, we construct multiple ODE systems by varying the value of $g^{}$ and run the system until it reaches an equilibrium; a line search algorithm is then applied to find an optimal solution that minimizes the attack and mortality rates. Due to the complexity and the nonlinearity of the ODE systems, each instance of switch trigger optimization requires about 6000 CPU minutes to solve. We develop a fast heuristic algorithm that returns near optimal solutions within 800 CPU seconds.

In addition to assigning each risk group its own disease compartments, the modeling framework is highly flexible and can be extended in multiple ways. Specifically, it can incorporate any types of interventions and resource requirements and analyze the overall effectiveness and tradeoffs. It can accept various POD layout configurations and any characteristics and structure of disease propagation. To optimize POD vaccine distribution operations, we leverage the computational capability and features of RealOpt-POD, a large-scale modeling and optimization decision-support system for public health emergency response developed at the Centers for Disease Control and Prevention [39,40,41,53]. We also use the agent-based simulation module in RealOpt to trace the behavior and disease status of each individual inside the PODs. This allows comparison and validation of results obtained for this new vaccination allocation-distribution system.

## 3. Results

We analyze COVID-19 vaccine prioritization for the Greater Houston metropolitan area with 7.0 million population, among which 16.6% are high-risk (5.4% healthcare workers, 10.2% 65+ years-old, and 1.0% patients under 65 with high-risk health conditions) [54,55]. A total of 2.8% are immune from previous infection [56]. Seventy-five PODs are set up across the city to vaccinate the citizens efficiently. Two doses will be administered per individual, given 21–30 days apart. The efficacy is estimated to be 70–95% [57]. Since it takes so long to induce immunity (Pfizer: 21 days + 7 to 14 days), protective measures including use of facemasks, social distancing, strategic testing, isolation, and quarantine continue during the periods when vaccines are dispensed. At the time of our analysis, the estimated basic reproduction number in Houston is $R_{0}=1.2$ with 0.5% active infectious population [58].

### 3.1. Vaccine Level Versus Switch Trigger

We first report the optimal switch triggers for a mixed vaccination strategy under five vaccine supply scenarios: 10%, 20%, 30%, 40% and 50% population coverage with no supply interruption. Figure 4a contrasts the results of the overall attack rate and mortality rate against high-risk group coverage. The optimal switch triggers, *g**, and the associated high-risk group population coverage under each supply scenario, are marked. Observe that when the vaccine supply is greater than 10%, if all high-risk individuals are prioritized to receive vaccine first, the attack rate and mortality rate are higher than when the optimal switch trigger is used. The higher the vaccine supply level, the earlier the vaccine is optimally open to the general population. Analyzing the 50% vaccine supply (green curves), the full prioritized policy results in a 4-fold increase in infection and a 3.2-fold increase in mortality over those from the optimal switch trigger (10.54% attack rate and 0.29% mortality rate versus 2.13% and 0.069%, respectively). We remark that a 1% reduction in attack rate corresponds to 70,000 fewer infectious individuals, thus the reduction amounts to 588,700 infections and 15,470 deaths. This demonstrates the importance of adopting the optimal switch trigger mixed strategy. In addition, the curvatures around the optimal solutions for all five scenarios are rather flat (maximum), showing that the optimal switch trigger is robust, and can be readily translated into practical implementation with a large fault tolerance. During an emergency, the inventory data of vaccines, the calculation of proportions of high-risk populations, and the record of administrated vaccines may not be accurate. Thus, the robustness in our solution is critical and desirable.

Now consider an alternative scenario where vaccine becomes available intermittently, in batches over time. Assume the batched vaccine supply arrives in an increment of 10% with 30 days apart between each shipment: Day 0, Day 30, Day 60. We can clearly observe the inferior results when vaccine arrives in batches with interruption. Although eventually 30% of vaccines are available, the resulting attack rate (black solid curve) is inferior to a 20% uninterrupted vaccine supply (yellow solid curve), underscoring the difficult competition of racing to vaccinate against the continuous disease spread. Hence timely supply and vaccination is a must.

Under the basic reproduction number of $R_{0}=1.2$, the optimal mixed strategy is superior to both full nonprioritized and full prioritized strategies. The attack rates of the five scenarios under full nonprioritized policy are 36.79%, 23.82%, 11.95%, 4.84%, and 2.54% respectively; whereas the attack rates under the full prioritized policy are 29.02% for 10% vaccine supply, and converge close to 10.54% to 12.95%, when vaccine supply is above 20%. The attack rates for the optimal mixed strategy are 29.02%, 11.04%, 5.44%, 3.14%, and 2.13%, respectively.

Figure 4b shows the increase in attack rate and mortality rate with respect to the optimal mixed strategies for the full prioritized and full nonprioritized strategies as the vaccine inventory increases. We can observe that as more vaccines are available, there is little incentive to prioritize and that a large portion of the vaccine should be open to the public quickly. Even at 20% supply, we see a clear benefit in adopting the optimal switch trigger for the best population protection.

Contrasting the reduction in infection and mortality (the solid curves versus the dashed curves), for the high-risk individuals, as vaccines are open to the public, mortality reduction is less than infection reduction since high-risk individuals have higher mortality rate (navy blue curves). Likewise, mortality reduction is slightly higher than infection reduction when vaccine is dedicated to the high-risk (pink curves), since the general public has lower mortality. The optimal mixed strategy offers a tradeoff balance between the two objectives: minimizing total infection versus total deaths.

Figure 5 shows the overall infection during the COVID-19 outbreak under three vaccination strategies: optimal mixed strategy, full prioritized, and full non-prioritized, when the vaccine supply level is 30% and $R_{0}=1.2$. Individual-based stochastic simulation (dotted-dashed) from RealOpt is used to verify the output of the ODE systems (solid curves). Both methods suggest the same trend: the optimal mixed strategy can significantly reduce the attack rate and mortality throughout the outbreak and leads to timely disease containment. Focusing on the 30% non-interrupting supply (Figure 5a), with full prioritized strategy (pink curve), the attack rate at containment is 11.51%, and the outbreak starts to contain at around Day 256. The optimal mixed strategy (purple curve) achieves an overall attack rate of 5.44% with time of containment at Day 185, a 27% timeline improvement and 111% infection reduction compared to the full prioritized strategy. Note that the optimal mixed strategy from the batched supply (black curve) results in an attack rate of 10.45% at containment, a 92% increase over the non-interrupted supply (Figure 5b).

These figures show the best possible overall infection outcome based on various vaccination strategies. Under the optimal mixed strategy, we can see a clear march to a plateau on the attack rate within 3 to 4 months, while it takes 5–7 months under the other two strategies. We caution that infection surges could occur due to variants and other external human and social factors including gatherings during holidays and major life events [56,59].

### 3.2. Delay in Supply and Vaccination, and Dispensing Operations

Figure 6 depicts the effect of delay in supply of vaccines or vaccination. Looking at the 30% vaccine supply curves (purple), when there is no delay (solid purple curve), the peak of the daily disease prevalence occurs at Day 22 with 0.579% maximum and starts to reduce afterward. When the vaccine supply or vaccination is delayed by 3 weeks (dotted purple curve), the number of infectious individuals reaches a peak by Day 42 with a maximum prevalence of 1.435%. The longer the delay, the later the peak appears since the spread of disease is not effectively controlled before the arrival of vaccines and the administering of the vaccine, and thus the infectious population keeps growing. Besides the longer time needed to achieve containment, the delay also significantly increases the cumulative infections and deaths. This underscores the importance of both the vaccine availability and the ability to administer them without delay to ensure effective disease containment.

We observe that infections continue to rise for a while after the first vaccination. Depending on the vaccination effort, the time before reduction begins ranges from 28 days to 70 days. This latency phenomenon partially reflects the time required for immunity to kick in, and the real-life compliance of masking and social distancing. Vaccinated individuals may feel protected and become lax in compliance. Our findings appear to reflect the on-the-ground real situation well [56,59]. We note that Texas had a state mask mandate in place between the period 3 July 2020 to 10 March 2021, during which compliance rates hovered between 80% to 90%.

To quantify the impact of delay of initial vaccine supply in containing the outbreak, we contrast the overall attack rate and mortality rate versus the vaccine supply level under a time delay of 1 to 3 weeks. Figure 7 shows a clear setback in containment across all levels due to delay. A delay of one week increases the attack rates by 2.0% to as much as 58.0% for vaccine supply levels ranging from 10% to 50%. Likewise, mortality rates increase by 2.5% to as much as 62.2%. Three weeks delay seriously dampens the response effort with both the attack rate and mortality rate increasing by over 200% for vaccine levels greater than 20%. Our analysis again underscores the importance of early vaccine availability and the need to vaccinate without any delay. The results also drill into the necessity of vaccine supply without interruption and the precious time loss to disease spread. Although the batched supply has a total of 30%, the overall impact is far inferior due to its interrupted arrival schedule.

Figure 8 shows the attack and mortality rates at containment under the optimal mixed vaccination strategies against dispensing efficiency at the vaccine clinics. When the throughput efficiency is higher than 60%, the impact on effective outbreak control is negligible. However, the attack rate and mortality rate increase significantly when the dispensing efficiency drops below 40%. Therefore, maintaining a high dispensing efficiency at all vaccine clinics is crucial and essential in accelerating disease containment. With available vaccines, POD managers must optimally allocate resources to achieve maximal throughput possible to ensure timely disease containment.

Individuals visiting the vaccine clinics or PODs may be pre-symptomatic infectious or asymptomatic, and hence may facilitate disease spread. Therefore, proper contact-tracing and screening should be carried out, and social distancing should be observed, as large crowds at vaccine clinics and PODs increase risks of infection.

### 3.3. Severity of the Pandemic and Initial Prevalence

Table 1 contrasts the optimal switch trigger for mixed strategy under different combinations of vaccine supply level and basic reproduction number $R_{0}$. When *R*_0_ increases, the optimal switch trigger for all supply levels drops, indicating an earlier switch to the nonprioritized strategy. This observation is intuitive since the more severe the pandemic is, the earlier the strategy protects all populations instead of only the high-risk population. This is because a high basic reproduction number will dominate the effect of high infectivity and mortality associated with high-risk population. The results also suggest that the overall attack rate and mortality of the optimal mixed strategy is much lower compared to the other two strategies, cutting as much as half the infections throughout the COVID-19 outbreak in some cases. In addition, we observe that by using the optimal mixed strategy, the attack rate can be superior to that of non-optimal strategies with higher vaccine supply levels (for example, for all $R_{0},\;$ an optimal mixed strategy at 30% vaccine inventory offers better overall protection than a 40% supply full prioritized strategy). This underscores the significance of optimal control in vaccine allocation and distribution for disease containment.

Figure 9 shows the optimal switch trigger of mixed strategy with respect to different basic reproduction number, initial percentage of infection, and the level of vaccine supply. Closer analysis shows that the optimal switch trigger drops below 100% in all combinations of $R_{0}$ and initial infectious population when the vaccine supply level exceeds 12%. This conclusion can simplify the determination of optimal switch trigger in real emergencies: when the level of vaccine supply is below 12%, all should be dispensed to high-risk groups; the switch between strategies needs to be considered only when the vaccine supply exceeds this threshold. The results suggest that the performance of the optimal mixed strategy is quite robust against model parameters. This property is particularly useful in promoting and adopting the use of optimal mixed strategy: when combating various types of infectious disease outbreaks (COVID-19 or others), even if the estimates of $R_{0}$ and initial infection may not be accurate, it does not have a significant impact in the optimal switch triggers.

### 3.4. Vaccine Efficacy and Immunity, and Diagnostic Accuracy

The rapid design of COVID-19 mRNA vaccines offers great promise to contain the disease and allows citizens to get back to a new normal life. However, numerous uncertainties exist in the efficacy of the vaccine, duration of immunity, the level of protection in transmission reduction, and more importantly, their effectiveness against new variants. As the more contagious variants from United Kingdom [60] are raging in United Kingdom and Europe and moving rapidly through the United States, public health and infectious disease leaders are racing to investigate and understand the sustained protection offered by the vaccines. There are also other variants, including the South African, Brazilian, and Danish that are circulating around different parts of the world, including the United States [61]. While the ultimate effectiveness and risks and benefits of vaccines can only be ascertained through public health investigation, analytic investigations can shed insights into potential risks and outcomes that we should anticipate. Such information can advise the public regarding vigilance in social distancing and use of facemasks for continued protection.

Figure 10 depicts how the overall attack and mortality rates change with respect to vaccine efficacy. Currently, both mRNA vaccines offer over 90% efficacy, which results in excellent overall attack rates, ranging from 5.44%, 11.04%, to 29.02% for vaccine supplies of 30%, 20% and 10%, respectively. As the efficacy drops to 50%, we see a rapid uptake of infection, increasing from 34% for the 10% supply to over 316% for 30% vaccine supply. This underscores the urgency in developing vaccine boosters to fight against the more contagious Brazil, United Kingdom, and South Africa variants. This also showcases the vulnerability of our population and the critical importance to continue non-invasive non-pharmaceutical interventions to minimize any potential disease escalation. Rapid disease containment of the current outbreak is of paramount importance.

We assume the mass vaccination plan is to offer vaccine to everyone who has no prior COVID-19 infection, where prior infection is confirmed through a positive molecular RT-PCR test. RT-PCR is extremely sensitive and detects viral nucleic acid and reflects acute infection or perhaps residual RNA post-infection. Numerous studies have been conducted to gauge the accuracy of the tests, with varying false negative rates depending on the day of test with respect to timing of infection [62]. We desire to understand how diagnostic accuracy may impact the overall effectiveness of the vaccine strategies.

Figure 11 depicts the optimal switch trigger with respect to the diagnostic accuracy of false negatives and false positives ranging from 0% to 10%. False negative individuals who have had COVID-19 will still be vaccinated. While this may use up valuable vaccine resources, false negatives have only a slight impact on the number of infections and mortalities, and do not significantly change the corresponding optimal switch trigger.

On the other hand, the optimal switch trigger increases as the diagnostic false positive rate decreases. When the false positive rate is 10% (i.e., diagnostic accuracy is 90%), the optimal triggers for the three vaccine supplies are 69.69%, 44.27%, and 41.19%, respectively, as opposed to 78.84%, 50.32%, and 48.55% when the accuracy is 100% (no false positives). Since more healthy/susceptible individuals are diagnosed as infectious and hence do not receive vaccines, fewer vaccines will be dispensed, and the time of switching to the non-prioritized strategy will be earlier.

## 4. Discussion

In this study, we propose a general-purpose modeling framework for infectious disease outbreaks with human interventions. We focus on the COVID-19 outbreaks scenario and study the effect of different vaccination strategies due to a limited supply of vaccine on overall attack and mortality rates. We propose the idea of an optimal switch trigger, which represents the percentage of vaccine given to the high-risk population before switching to a non-prioritized vaccination strategy. We integrate the multi-purpose compartmental disease propagation model with the human behavior network, the (hospital) resource logistics model, and the stochastic vaccine queueing model, into a single computational platform and then develop numerical optimization on that framework. To ensure realism, we use vaccination data from our previous mass vaccination time-motion studies to populate the vaccine clinic model parameters and determine human behavior within the model. Empirical results are obtained for hundreds of cities and counties in the United States. They follow a consistent pattern; and for brevity, herein we only report detailed analysis for Greater Houston. Sensitivity analyses are included to investigate the impact of various perturbations in implementing the optimal mixed vaccination strategy, including perturbations in vaccine efficacy, delay of vaccine supply, dispensing efficiency, and diagnostic accuracy.

A full prioritized vaccination strategy is one where all vaccines are distributed first to high-risk groups, whereas a full nonprioritized is one where vaccines are administrated to everyone with no special priority given to high-risk groups. A mixed strategy is a hybrid approach that first prioritizes the vaccines to cover some of the high-risk population, and switches to the general public at a later time. Our analyses suggest that the optimal mixed vaccination strategy outperforms both the full prioritized and the full non-prioritized strategies in minimizing the overall attack and mortality rates under different combinations of model parameters and availability of vaccines (Figure 4a, Table 1). When the vaccine supply is very limited, all vaccines should be dispensed to high-risk groups for best containment results. But when the vaccine supply exceeds a threshold, a switch in the dispensing strategy can be favorable and result in lower attack and mortality rates (Figure 9). This threshold hovers between 12–16% among the hundreds of cities that we have analyzed. Delay of vaccine supply or vaccinations can have a significant impact on the trend of the disease outbreak and may result in an increase in the cumulative infectious population (Figure 7); thus, early dispensing operations and availability of vaccines are crucial in containing disease outbreaks. The result of implementing the optimal mixed strategy can also be influenced by various factors, including the operational efficiency at vaccine clinics/PODs, vaccine efficacy, and diagnostic accuracy (Figure 8, Figure 10 and Figure 11).

The optimal switch trigger is robust and is achieved with a large fault tolerance, thus forgiving to the many uncertainties in inventory supply accounting, risk group assessment, and exact administered doses. Our analysis reveals that public health leaders should ensure vaccines are dispensed at high efficiency and safeguard intra-facility disease spread by means of smart POD layout design that provides short queues and wait times. Use of electronic health records (EHR), use of electronic scanning technologies [63], optimal design of POD layout, and optimal allocation of healthcare workers, are important to facilitate achieving the best containment results [39,40,41].

Given the emergence of treatment-resisting and vaccine-evading variants, such as the South African and Brazilian variants, it is desirable to reduce spreading as quickly as possible so as to lower the chances for further mutations to evolve and to reduce the excessive healthcare burden (Figure 10). Further analysis on vaccine efficacy and virus variants are underway.

The high-risk partition presented herein represents the optimal risk categories that are most impactful for rapid disease containment. Comprehensive analysis of how inclusion of different risk-groups affects the optimal switch trigger and the containment timeline will be detailed in a subsequent study.

To the best of our knowledge, this is the first general-purpose computational model designed for guiding intervention efforts to contain infectious disease outbreaks that combines biological characteristics, human behavior, hospital resources and intervention operations logistics, into a single platform (Figure 12) [64]. The framework accounts for disease progression and transmission (light blue) that vary according to the specific pathogen. At the same time, it incorporates human behavior, social and environmental influences, and risk factors (light beige) to model heterogeneous transmission dynamics on how disease spread takes hold. Interventions, pharmaceutical or non-pharmaceutical, in combination or alone, play crucial roles in containing the disease. For example, behavioral changes such as handwashing and use of facemasks can have significant positive impact in combating contagious infectious disease, and they are both human/social behavior change as well as a means of non-pharmaceutical intervention. The model was first utilized in evaluating timelines for non-pharmaceutical interventions, including school closure, business telecommuting, social distancing, and face masking in March 2020; providing recommendations to state and federal policy makers.

This modeling framework accommodates populations with different risks by assigning each risk group its own disease compartments. It is noteworthy that traditional ODE models have mostly been inadequate for COVID-19, largely because what drives the rate of transmission is not only the contact rate of the most active individual spreader but the formation of groups.

The system facilitates the analysis of vaccine security, exploring vaccine benefits and risks and tradeoffs and how they manifest across different risk groups and populations. Specifically, different groups can have their own disease compartment modules, and each module can be coupled with the vaccination process flow shown in Figure 3, which can include risks and adverse effects along the pathways. The findings can have serious implications in decision making for both public health leaders and the public. In a CDC report, data shows that out of 13.7 million Americans vaccinated during December 2020 and January 2021 side effects were higher among women [65]. Specifically, “Headache (22.4%), fatigue (16.5%), and dizziness (16.5%) were the most frequently reported symptoms after vaccination with either vaccine. Sixty-two reports of anaphylaxis have been confirmed, 46 (74.2%) after receipt of the Pfizer-BioNTech vaccine and 16 (25.8%) after receipt of the Moderna vaccine.”

Investigations of vaccine adverse effects of cases of rare blood clots resulting from the AstraZeneca vaccine are ongoing by the European Medicines Agency, and Medicines and Healthcare products Regulatory Agency [66,67]. At the time of this writing, CDC and FDA have put a pause on the Johnson and Johnson vaccine.

It is critical to have a better understanding of vaccine risks and benefits. It affects mass vaccination campaigns, public confidence, and vaccine hesitancy. We are currently investigating these issues and will report our findings in a future paper.

This modeling framework is highly flexible: it is not constrained by a single type of disease or intervention method. To model different types of diseases—e.g., vector-borne diseases such as Zika, dengue and yellow fever, or food-borne diseases like cholera—decision makers only need to plug in the corresponding structure for the compartmental disease model along with estimates of parameters based on biological information and then use the modeling framework and computational engine to derive optimal interventions, such as the switch trigger for the best mixed vaccination strategy. In addition, our definition of optimal switch trigger integrates not only the property of disease propagation itself but also the vaccine supply and logistics in actual vaccine dispensing operations. Other important factors that impact the decision-making process are also highlighted in the sensitivity analysis, making the framework a practical and effective tool to use in real emergencies.

The ordinary differential equation (ODE) system, the human connectivity, and the resource and logistics queueing-based modeling framework are computationally intense already; finding the optimal switch trigger adds to the complexity. Our result suggests that the agent-based simulation model for disease propagation has good performance similar to that of the ODE-based model; but it remains computationally challenging. We develop efficient numerical strategies to improve the time needed for obtaining the optimal switch trigger for the agent-based simulation model (ABM) in RealOpt-ABM©, and we are currently integrating the efficient solver for the approximated solution of the ODE system into a general-purpose digital disease surveillance and response system, RealOpt-ASSURE©. RealOpt-ASSURE allows users to understand the disease dynamics and the effectiveness of different intervention methods, and it assists decision-makers in choosing the best combination of strategies via simulations of possible trends of disease outbreak development.

With an increasing danger of global disease outbreaks and insufficient supply or even unavailability of vaccines, it remains a challenge to determine the best way to protect more people despite limited resources. The SARS-CoV-2 virus and COVID-19 present grave challenges in determining the best public health pandemic response. In addition to analyzing vaccine immunogenicity prediction on an individual level for optimal utilization of vaccines, we tackle this problem by analyzing the strategies for allocation and dispensing. One advantage of our approach is that decision-makers only need to know an estimate of the overall vaccine efficacy to derive the optimal switch trigger. In addition, sensitivity analyses indicate that the optimal switch trigger is highly robust against multiple parameters. A near-optimal result can still be achieved and useful in practice even when accurate estimations of model parameters are not immediately available. This makes our modeling framework attractive in practice as it empowers government and public health leaders to take risk-informed rapid actions in emergencies.

## Figures and Tables

**Figure 1 vaccines-09-00506-f001:**
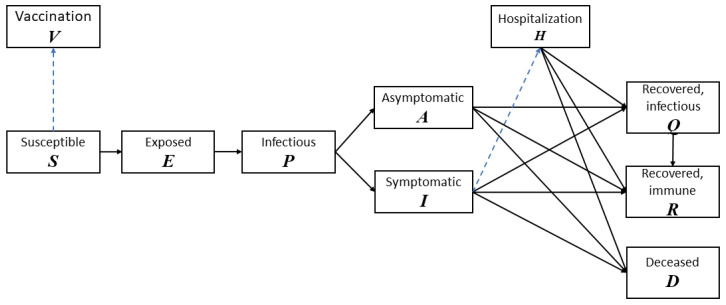
The stage transition diagram of our COVID-19 model. Solid lines are transitions associated with new infections and disease propagation. Dashed lines are transitions associated with vaccination and treatment.

**Figure 2 vaccines-09-00506-f002:**
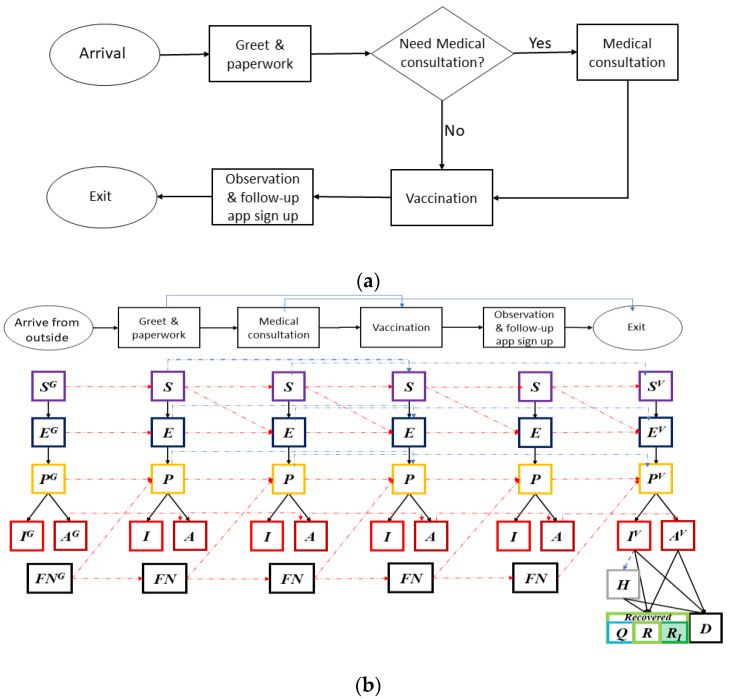
(**a**) A process flowchart of the COVID-19 point-of-dispensing (POD) operations. The service time at each station is fitted with real-life time-motion studies from multiple vaccination events. (**b**) Disease stages of individuals within each station of the POD, their interplay and progression dynamics. A susceptible individual at a station may become exposed and move to “Exposed” disease stage in the next station. False negative (FN) individuals may have recovered from COVID-19; or a newly FN individual can become infectious at some point inside the POD. Recovered individuals can be recovered with immunity (*R_I_*) or recovered without immunity (*R*).

**Figure 3 vaccines-09-00506-f003:**
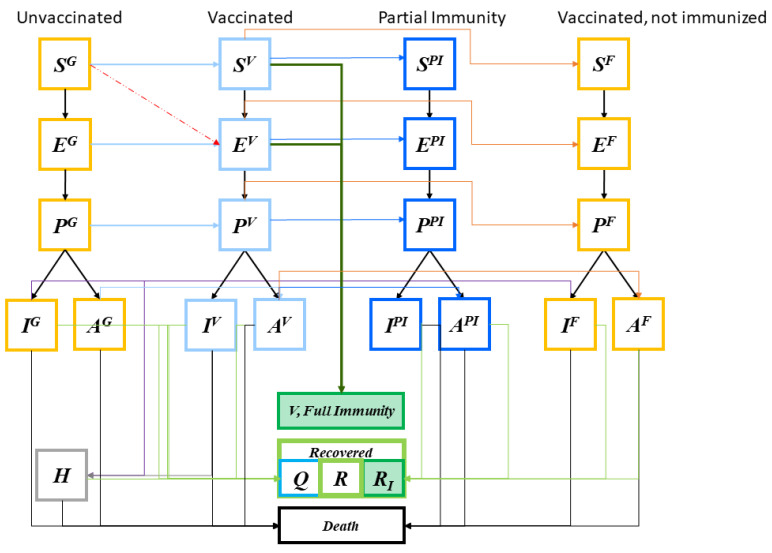
Immunity and disease stages of individuals upon vaccination. Individuals may become exposed to the disease during the vaccination process (red arrow). Once vaccinated, it typically takes a few weeks for the body to establish immunity against the virus. A person could still get COVID-19 just after vaccination (the light-blue disease progression). Some vaccinated people may develop only partial immunity and may contract disease still, although mostly with only mild or no symptoms at all (dark blue disease progression). A small percentage may fail to develop immunity.

**Figure 4 vaccines-09-00506-f004:**
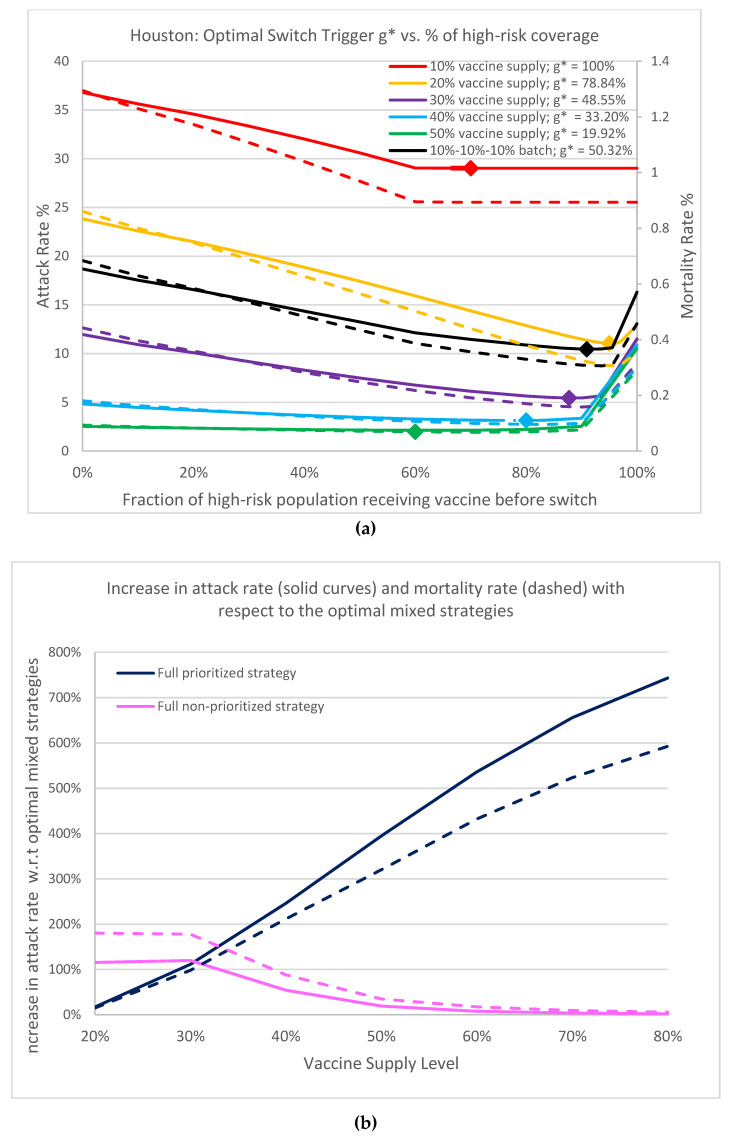
(**a**) Plot of overall attack rate and mortality rate (evaluated on Day 360) as a function of percentage of high-risk coverage. $R_{0}=1.2$. Red, yellow, purple, blue and green curves correspond to, respectively, 10%, 20%, 30%, 40% and 50% uninterrupted vaccine supply inventory. The black curves show the batched 30% vaccine supply with delay: 10% arrived on Day 0, Day 30, and Day 60. Solid curves denote attack rates, dashed denote mortality rates. Points marked on the curves indicate the associated optimal switch trigger. (**b**) This graph shows the rapid increase in attack rates (solid curves) and mortality rates (dashed curves) for full prioritized strategies with respect to the optimal mixed strategies as more vaccine becomes available. It also shows that there is more urgency to open up more vaccines to the general public.

**Figure 5 vaccines-09-00506-f005:**
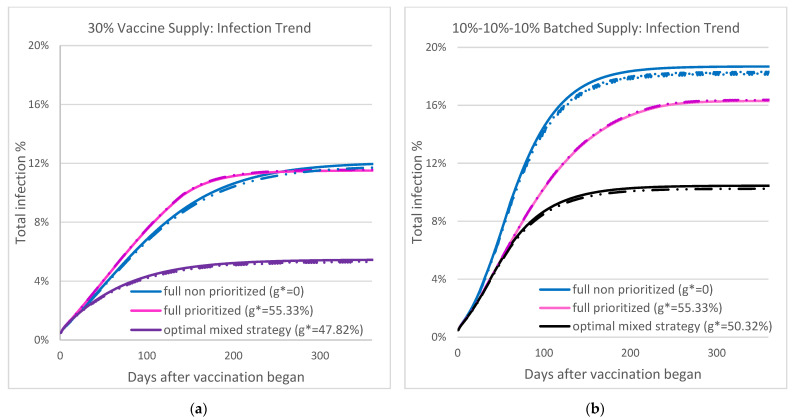
The two figures show overall infection under three vaccination strategies—optimal mixed strategy, full prioritized, and full non-prioritized—when the vaccine supply level is 30%, *R*_0_ = 1.2, and initial infection is 0.5%. The solid curves are results from the ODE-based system and the dotted-dashed curves are results from RealOpt agent-based simulation. (**a**) Results from non-interrupting supply. (**b**) Results from batched supply. Note the optimal mixed strategy from the batched supply (b. black curves) results in 92% increase in attack rate over the non-interrupted supply (a. purple curves).

**Figure 6 vaccines-09-00506-f006:**
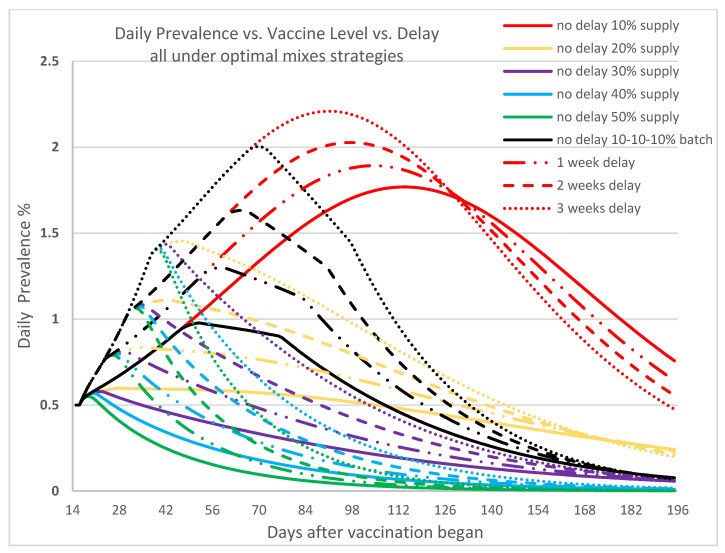
In this figure we show daily prevalence of COVID-19, resulting from various vaccine supply levels when vaccination begins on time, or with delays of one week, two weeks, and three weeks, and is dispensed according to the associated optimal mixed strategy. Here, $R_{0}$ = 1.2 and the initial infected population is 0.5%. The *x*-axis represents the number of days since vaccination began. The colors correspond to the vaccine levels, the line types correspond to the delay timeline.

**Figure 7 vaccines-09-00506-f007:**
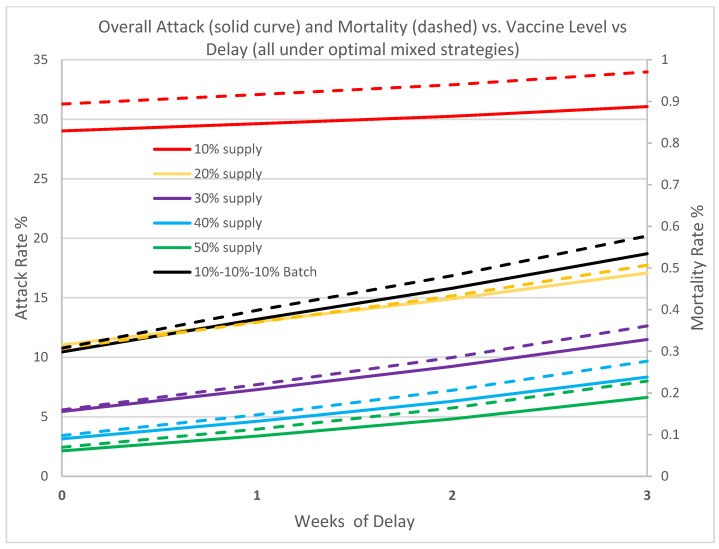
In this figure we contrast the overall attack rate and mortality rate (Day 0 to Day 180) associated with six different vaccine supply/delay scenarios, all operating under the optimal switch triggers. The solid lines denote attack rate, and the dashed lines denote the mortality rate. $R_{0}$ = 1.2, initial infection is 0.5%. Note the diminished effect of the batched supply (black).

**Figure 8 vaccines-09-00506-f008:**
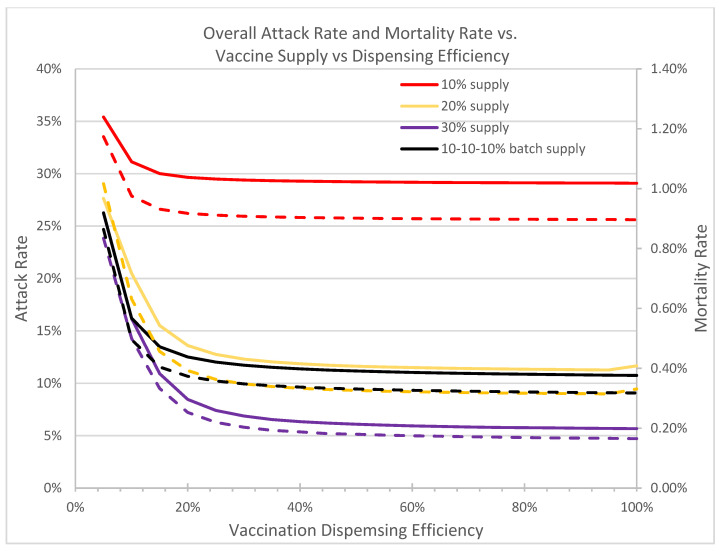
In this figure we compare the overall attack and mortality rates under the optimal switch triggers against different dispensing throughput efficiency levels at the vaccine clinics. $R_{0}$ = 1.2, initial infection is 0.5%. Solid curves denote attack rates; dotted curves denote mortality rates.

**Figure 9 vaccines-09-00506-f009:**
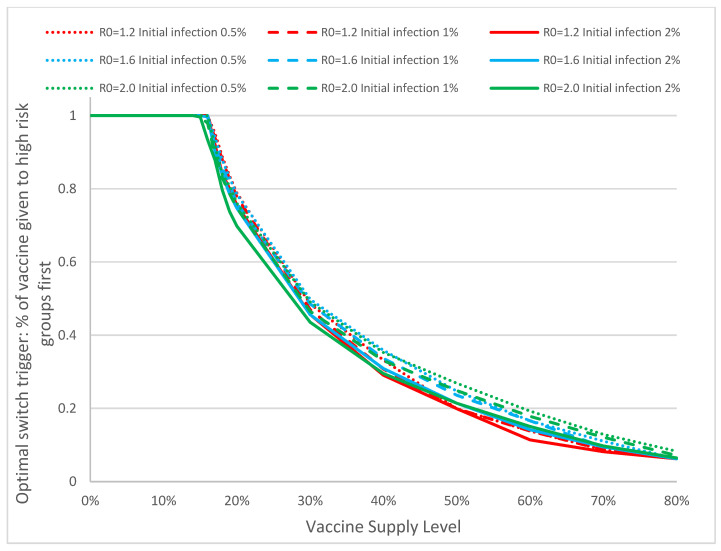
In this panel we show the optimal switch trigger (percentage of vaccine dispensed to the high-risk groups) against the vaccine supply levels in the optimal mixed strategy for different combinations of basic reproduction number and percentage of initial infectious population. Analyzing the values closely, we see that when the vaccine supply covers more than 12% of the entire population, not all individuals in the high-risk group need to be vaccinated before switching to non-prioritized strategy.

**Figure 10 vaccines-09-00506-f010:**
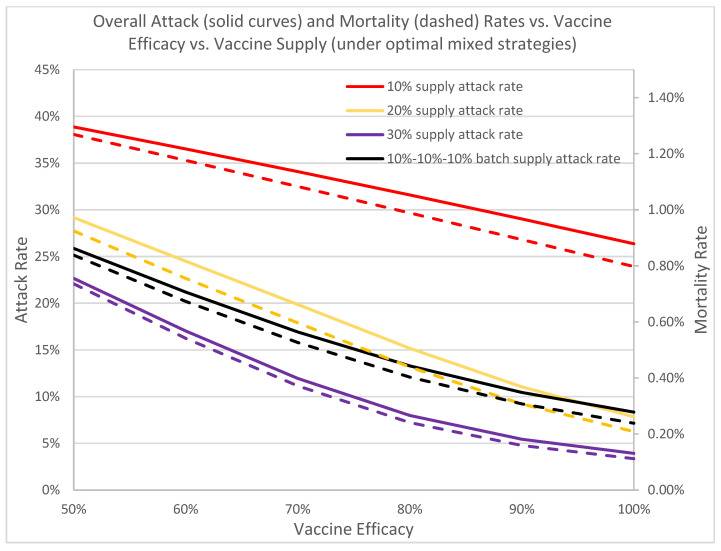
In this figure we depict the overall protection of the vaccine with respect to its efficacy across different levels of vaccine supply. The rapid decreases in attack and mortality rates across all vaccine supply levels as vaccine efficacy increases underscores the importance of rapid vaccination to contain the current outbreak and the development of booster shots to safeguard the public from new variants. Solid curves denote attack rates; dotted curves denote mortality rates.

**Figure 11 vaccines-09-00506-f011:**
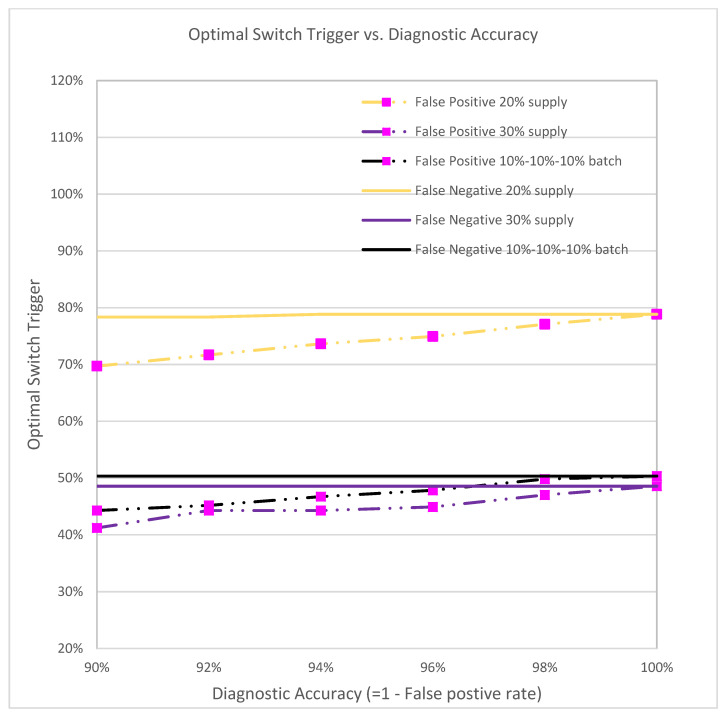
The optimal switch trigger under different diagnostic accuracy rates.

**Figure 12 vaccines-09-00506-f012:**
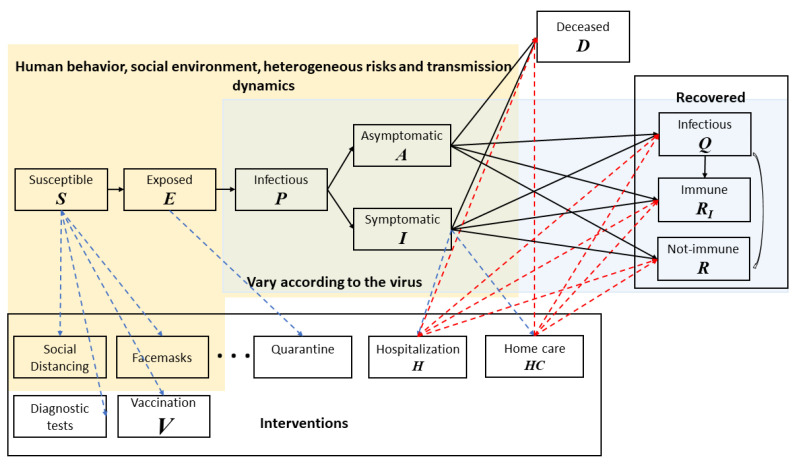
The biological-behavior-intervention informatics computational framework. The solid line arrows represent the natural disease progression. The blue dotted line arrows represent paths individuals take on an intervention; it can be via behavior or social environmental changes; or due to available resources (e.g., diagnostic tests, hospital beds). The red dotted line arrows reflect pathways of patient conditions upon receiving intervention. The figure includes some example interventions and pathways.

**Table 1 vaccines-09-00506-t001:** Optimal switch trigger for mixed strategy under different vaccine supply levels and basic reproduction numbers, the associated attack rates achieved; and their comparison to the full prioritized and the full nonprioritized strategies.

*R* _0_	Vaccine Supply	Optimal Mixed Strategy		% Increase in Attack Rate	
		Switch Trigger	Attack Rate	Full Prioritized	Full Nonprioritized
1.2	20%	78.84%	11.05%	+17.18%	+115.63%
	30%	48.55%	5.44%	+111.49%	+119.68%
	40%	33.20%	3.14%	+246.08%	+54.17%
1.6	20%	78.48%	39.98%	+11.18%	+18.69%
	30%	49.80%	26.89%	+64.39%	+29.19%
	40%	35.86%	14.71%	+199.31%	+49.82%
2.0	20%	76.03%	56.84%	+7.98%	+6.46%
	30%	49.09%	44.27%	+38.65%	+9.33%
	40%	35.25%	31.38%	+95.59%	+14.76%

## Data Availability

All data used in this manuscript is obtained from public sources or from peer-reviewed literature (Table A1). COVID-19 data: New York Times Coronavirus (Covid-19) Data in the United States, https://github.com/nytimes/covid-19-data. Greater Houston data: https://houstonemergency.org/houston-covid-19-deaths/, https://www.houston.org/reopen-safely-dashboard (accessed on 17 June 2020), U.S. Census Bureau: https://www.census.gov/ U.S Bureau of Labor: https://www.bls.gov/ (accessed on 20 August 2020).

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
