# Peer review of "Strategies for Vaccine Prioritization and Mass Dispensing"

_vaccines, 2021, doi:10.3390/vaccines9050506_

Round 1

Reviewer 1 Report

The manuscript is scientifically sound and of interest for Vaccines; however, one minor comment remains necessary in the reference section. References should be consistent; authors used the full Journal name in some cases and abbreviated them in other cases. 

Author Response

The manuscript is scientifically sound and of interest for Vaccines; however, one minor comment remains necessary in the reference section. References should be consistent; authors used the full Journal name in some cases and abbreviated them in other cases. 

Response: Thank you for the comments.  We have updated the references to ensure consistency.

Reviewer 2 Report

The authors are to be thanked for their considerable efforts in performing this extensive study. This approach is relevant and will provide helpful information for the decision-makers regarding the COVID-19 vaccination. The methodology is sound and well described, and the main effort is about the sensitivity analysis for this modeling approach, which considerer multiple change scenarios to support the main results. 

There are only one central question and some minor suggestions for the authors to clarify or even explain the results' applicability, even some statements or modeling assumptions in the COVID-19 scenario and the discovery of new virus mutations or strains.

The central question is about the assumption that after the first vaccination dose the use of mask, social distancing will remain or even better. Still, according to IHME data (https://www.worldometers.info/coronavirus/#countries), there is an increase in new cases and deaths. Elders and their caregivers could increase the social mobility after the first dose or even the after the second without proper use of a mask or social distancing. What will this implication mean for the study results? What would affect with the mixed approach will remain stable, increase or decrease the outcomes?. 

It would be beneficial to calculate the number of individuals affected, and it would clarify the intervention's implications, as you referred to in lines 286-288. "We remark that a 1% re- 286 duction in attack rate corresponds to 70,000 fewer infectious individuals, demonstrating 287 the importance of adopting the optimal switch trigger mixed strategy. "

Figure 4. Numbers inside the graph saturate the graph, and the message, a table inside the figure could solve the problem. 

Figure 5. The modeling approach assumes that mobility will be the same for elders and caregivers, but there is an apparent effect of increasing social mobility related to epidemic control. December Holidays and Thanksgiving day were associated with the increasing number of cases 15 days after; as shown IHME by https://covid19.healthdata.org/united-states-of-america?view=total-deaths&tab=trend

In line 462, "or 30% vaccine supply! "replaced with "or 30% vaccine supply. "

Additionally, self-citations are not justified and cited. No journal or how to access the statement is needed. Citation number 52, 63, 73, 74, 75, and 76.

Author Response

The authors are to be thanked for their considerable efforts in performing this extensive study. This approach is relevant and will provide helpful information for the decision-makers regarding the COVID-19 vaccination. The methodology is sound and well described, and the main effort is about the sensitivity analysis for this modeling approach, which considerer multiple change scenarios to support the main results. 

There are only one central question and some minor suggestions for the authors to clarify or even explain the results' applicability, even some statements or modeling assumptions in the COVID-19 scenario and the discovery of new virus mutations or strains.

The central question is about the assumption that after the first vaccination dose the use of mask, social distancing will remain or even better. Still, according to IHME data (https://www.worldometers.info/coronavirus/#countries), there is an increase in new cases and deaths. Elders and their caregivers could increase the social mobility after the first dose or even the after the second without proper use of a mask or social distancing. What will this implication mean for the study results? What would affect with the mixed approach will remain stable, increase or decrease the outcomes?

Response: Thank you for the comments. Indeed, as shown in Figure 6, our analysis conducted during the period September to November 2020 and presented herein, reflected the current IHME data. The infection increases for a while after the first vaccination, before it reduces. Depending on the vaccination effort, the delay effect can be 28 days to 70 days after vaccination begins. Since most states have no mandate for masking, masking compliance has hovered between 80% to 90%.  The mixed approach remains stable. The overall attack rates in Figure 6 represent the best possible outcome. Infections will increase as a result of variants and delays in vaccination due to operations logistics as analyzed, or difficulty in registering for the under-served or disfranchised population or public hesitancy. In addition, lax compliance with masing and social distancing will increase infections.  We added the following comments (lines 377-384) and include HIME in the citation.

Line: 377-384: “We observe that infections continue to rise for a while after the first vaccination. Depending on the vaccination effort, the time before reduction begins ranges from 28 days to 70 days. This latency phenomenon partially reflects the time required for immunity to kick in, and the real-life compliance of masking and social distancing. Vaccinated individuals may feel protected and become lax in compliance. Our findings appear to reflect the on-the-ground real situation well [56,59]. We note that Texas had a state mask mandate in place between the period July 3 2020 to March 10 2021, during which compliance rates hovered between 80% to 90%.”

It would be beneficial to calculate the number of individuals affected, and it would clarify the intervention's implications, as you referred to in lines 286-288. "We remark that a 1% reduction in attack rate corresponds to 70,000 fewer infectious individuals, demonstrating the importance of adopting the optimal switch trigger mixed strategy.

Response:  We added a sentence in line 289-290. “thus the reduction amounts to 588,700 infections and 15,470 deaths.”.

Figure 4. Numbers inside the graph saturate the graph, and the message, a table inside the figure could solve the problem. 

Response:  Thank you for the suggestion. We updated Figure 4 as suggested.

Figure 5. The modeling approach assumes that mobility will be the same for elders and caregivers, but there is an apparent effect of increasing social mobility related to epidemic control. December Holidays and Thanksgiving day were associated with the increasing number of cases 15 days after; as shown IHME by https://covid19.healthdata.org/united-states-of-america?view=total-deaths&tab=trend. 

Response: Thank you for the comments. We certainly hope that COVID-19 infection will plateau before 2021 Thanksgiving and Christmas in the United States. Figure 5 shows the best possible overall infection outcome based on various vaccination strategies.  We keep all other parameters the same across the different approaches to get a fair comparison. We understand there can be infection surges during Thanksgiving and Christmas, which we have analyzed in our previous policy work in June 2020. We add these and the IHME citation on lines 360-360 to acknowledge these facts.

Line 355-360: “These figures show the best possible overall infection outcome based on various vaccination strategies. Under the optimal mixed strategy, we can see a clear march to a plateau on the attack rate within 3 to 4 months, while it takes 5-7 months under the other two strategies. We caution that infection surges could occur due to variants and other external human and social factors including gatherings during holidays and major life events [56,59].”  

 In line 462, "or 30% vaccine supply! "replaced with "or 30% vaccine supply. "

Response: Corrected.

Additionally, self-citations are not justified and cited. No journal or how to access the statement is needed. Citation number 52, 63, 73, 74, 75, and 76.

Response: 52, 73, 74, 75, 76 are removed. Preprint link is added for 63.

Reviewer 3 Report

The authors shows an interesting general-purpose computational model designed for guiding intervention efforts to contain infectious disease outbreaks in general and COVID-19 outbreak in particular. It combines biological characteristics, human behavior, hospital resources and intervention operations logistics, into a single platform. By using the overall attack rate and mortality rate as outcome metrics, the study support a mixed strategy or hybrid approach (that first prioritizes the vaccines to cover some of the high-risk population, and switches to the general public at a later time) instead of the other two strategies (full prioritized where all vaccines are distributed first to high-risk groups versus nonprioritized vaccination strategy where vaccines are administrated to everyone with no special priority given to high-risk groups). Lines 197-202: “elderly 65 years-old or above” Is one the two groups that are categorized as high-risk groups. However, it is supported on a 75 years old cut off point in the next sentence: “Elderly 75 year-old or above [46,47], and patients with associated health conditions are more vulnerable to infection and mortality [48]”. What about patients with high risk health conditions?; They must be a high-risk group…

In the Sensitivity analyses, perturbations in vaccine efficacy, delay of vaccine supply, dispensing efficiency, and diagnostic accuracy are included to investigate their impact in implementing the optimal mixed vaccination strategy. In addition, it is referred in the discussion section that further analysis on vaccine efficacy and virus variants are underway.

As a mayor comment, one important factor that impact the decision-making process is the security of available vaccines. This is, the potential benefits and potential harms of the vaccines, since the risk:benefit ratio varies between different people, and as prevalence of the virus changes. However, It seems not to be included in the model framework and no reference to this exits throughout the manuscript.

A highly contemporary example of this are the numbers of cases of the blood clot reactions provided by EMA* and MHRA** of the AstraZeneca vaccine. There are some very illustrative figures about this in the link below https://wintoncentre.maths.cam.ac.uk/news/communicating-potential-benefits-and-harms-astra-zeneca-covid-19-vaccine/?prm=ep-app

*European Medicines Agency information: https://www.ema.europa.eu/en/news/astrazenecas-covid-19-vaccine-ema-finds-possible-link-very-rare-cases-unusual-blood-clots-low-blood

**Medicines and Healthcare products Regulations Agency information: https://www.gov.uk/government/news/mhra-issues-new-advice-concluding-a-possible-link-between-covid-19-vaccine-astrazeneca-and-extremely-rare-unlikely-to-occur-blood-clots

Author Response

The authors shows an interesting general-purpose computational model designed for guiding intervention efforts to contain infectious disease outbreaks in general and COVID-19 outbreak in particular. It combines biological characteristics, human behavior, hospital resources and intervention operations logistics, into a single platform. By using the overall attack rate and mortality rate as outcome metrics, the study support a mixed strategy or hybrid approach (that first prioritizes the vaccines to cover some of the high-risk population, and switches to the general public at a later time) instead of the other two strategies (full prioritized where all vaccines are distributed first to high-risk groups versus nonprioritized vaccination strategy where vaccines are administrated to everyone with no special priority given to high-risk groups). Lines 197-202: “elderly 65 years-old or above” Is one the two groups that are categorized as high-risk groups. However, it is supported on a 75 years old cut off point in the next sentence: “Elderly 75 year-old or above [46,47], and patients with associated health conditions are more vulnerable to infection and mortality [48]”. What about patients with high risk health conditions?; They must be a high-risk group…

Response: Indeed for the stated 16.6% high-risk population, it includes healthcare workers (5.4%), elderly 65 years old or above (10.2%), and patients (under 65) with high-risk health conditions (1%). In the original manuscript, we put the latter 2 risk groups together since they are both health-condition related whereas healthcare workers’ risk is mostly work environment. In the revision, we have separated them into 3 groups with the associated percentage. Please see line: 201-202, 267-268.

In the Sensitivity analyses, perturbations in vaccine efficacy, delay of vaccine supply, dispensing efficiency, and diagnostic accuracy are included to investigate their impact in implementing the optimal mixed vaccination strategy. In addition, it is referred in the discussion section that further analysis on vaccine efficacy and virus variants are underway.

As a mayor comment, one important factor that impact the decision-making process is the security of available vaccines. This is, the potential benefits and potential harms of the vaccines, since the risk:benefit ratio varies between different people, and as prevalence of the virus changes. However, It seems not to be included in the model framework and no reference to this exits throughout the manuscript.

A highly contemporary example of this are the numbers of cases of the blood clot reactions provided by EMA* and MHRA** of the AstraZeneca vaccine. There are some very illustrative figures about this in the link below https://wintoncentre.maths.cam.ac.uk/news/communicating-potential-benefits-and-harms-astra-zeneca-covid-19-vaccine/?prm=ep-app 

*European Medicines Agency information: https://www.ema.europa.eu/en/news/astrazenecas-covid-19-vaccine-ema-finds-possible-link-very-rare-cases-unusual-blood-clots-low-blood

**Medicines and Healthcare products Regulations Agency information: https://www.gov.uk/government/news/mhra-issues-new-advice-concluding-a-possible-link-between-covid-19-vaccine-astrazeneca-and-extremely-rare-unlikely-to-occur-blood-clots

Response: Thanks for the important comment. Our model allows the analysis of vaccine security, the potential benefits and risks, and tradeoffs and how they manifest across different populations. Specifically, different groups can have their own disease compartment modules, and each module can be coupled with the vaccination process flow shown in Figure 3, which can include risks and adverse effects along the pathways. The findings will be reported in a future paper as more vaccine results are being reported now. At the time when we started the analysis and provided recommendations on how to roll out (June to November 2020), vaccines have been in clinical trials only. We have only begun the risk-benefit analysis two months ago, and are still waiting for more clinical results.

We added a paragraph (lines 604-622) with citations in the Discussion section to highlight these recent adverse effect findings and our ongoing analyses.

Round 2

Reviewer 2 Report

No comments in this version.

Reviewer 3 Report

The authors have appropiately responded to the comments, and the current version of the manuscript is substantially improved. It can be accepted for publication.